# A Multilevel Analysis of Factors Influencing School Bullying in 15-Year-Old Students

**DOI:** 10.3390/children10040653

**Published:** 2023-03-30

**Authors:** Yu-Jiao Wang, I-Hua Chen

**Affiliations:** 1School of Education Science, Liupanshui Normal University, Liupanshui 553004, China; 2Chinese Academy of Education Big Data, Qufu Normal University, Qufu 273100, China

**Keywords:** school bullying, influencing factors, multilevel study, student-level variables, school-level variables

## Abstract

Background: School bullying causes serious impacts on adolescents’ physical and mental health. Few studies have explored the various factors influencing bullying by combining different levels of data. Methods: Based on the database of four Chinese provinces and cities of the Program for International Student Assessment (PISA) in 2018, this study used a multilevel analysis model that combined school-level variables and student-level variables to explore the influencing factors of students being bullied. Results: Students’ gender, grade repetition, truancy and arriving late for class, economic, social, and cultural status (ESCS), teacher support, and parent support had significant explanatory power on school bullying on the student-level; on the school-level, school discipline atmosphere and competitive atmosphere among students had significant impacts on school bullying. Conclusions: Boys, students who have repeated grades, truancy and arriving late for class, and students with lower ESCS suffer from more severe school bullying. When developing school bullying interventions, teachers and parents should pay more attention to those students and provide more emotional support and encouragement to them. Meanwhile, students in schools with a lower discipline atmosphere and a higher level of competitive atmosphere experience greater levels of bullying, and schools should create more positive and friendly environments to prevent bullying events.

## 1. Introduction

In the 1980s, a young Norwegian boy committed suicide after suffering school bullying; since then, school bullying has begun to enter the field of researchers and has become an important research topic [1]. Nowadays, it is receiving more and more attention from many international organizations. Among the research topics related to school bullying, the primary focuses include the following: What characteristics cause individuals to be more likely to suffer from school bullying? Why are individuals with these characteristics easily bullied by others? What are the factors that cause bullying in schools? These issues have always been topics of great concern to researchers both nationally and worldwide.

Previous studies have shown that various types of factors affect students’ exposure to bullying, including individual characteristics, schools, and families [2,3,4,5,6,7]. Ruan examined the influencing factors of student suffering from school bullying through factor analysis and logistic regression [2]. The results showed that, from the cross-sectional dimension analysis, the ranking of factors was as follows: students’ individual characteristics, schools’ background features, and emotional support.

From a logistic regression of the longitudinal section, among the students’ background characteristics, boys were more likely to suffer from school bullying than girls; senior students were more likely to suffer from bullying than those in lower grades; and students with lower academic performance scores were more likely to suffer from school bullying than those with higher scores [2]. However, some studies have found different results. Regarding age, for example, Rigby and Slee found that younger children were more likely to experience bullying than older children [8]. With age increase, bullying tended to stop; the reason for this may be because individuals acquire more social skills that improve self-esteem [9].

In terms of school background characteristics, Ruan’s analysis showed that, compared with urban schools, students in rural schools suffered more school bullying; students in private schools suffered more school bullying than those in public schools; the more repeating students on a campus, the higher the proportion of school bullying, and the better the school discipline atmosphere, the fewer the bullying incidents [2]. Lastly, the class size, school size, and teacher–student ratio of a school’s background characteristics had no significant impact on students’ school bullying [2]. Contrary to Ruan’s results, however, Huang’s study found that school location (urban or rural) and school type (public or private) had no effect on students’ school bullying [5].

In addition, teacher support plays a very important role in school background characteristics. Effective teacher support greatly reduces the occurrence of school bullying, but if teachers treat students unfairly, it may increase the occurrence [5]. Regarding the home environments of school bullies and victims, children who perpetrated bullying reported that their parents did not exercise caring and supervisory functions, sometimes even neglecting them [7]. This is in contrast to the home environments of bullying victims, who had very close relationships with their parents and were, therefore, vulnerable to overprotection. In addition, Fu et al. pointed out that students from families with lower socioeconomic statuses were more likely to be victims of more severe types of bullying, as school is an integral indicator of social stratification [10]. Parental emotional support was an important family factor affecting students’ suffering from school bullying, and insufficient parental emotional support was an important reason why many young people suffered from school bullying and could not cope effectively [5].

In conclusion, it can be seen that school bullying is affected by various factors of individuals, families, and schools, but there have been some contradictions among past studies, such as age and school type, which may be related to sampling or research methods. In addition, few studies have explored the various factors influencing bullying by combining different levels of data. When discussing this topic, these influencing factors should be considered comprehensively, but different levels cannot be analyzed at the same level, which leads to analytical bias. When facing these data from different sources, a multilevel analysis method should be used for an accurate analysis.

The Program for International Student Assessment (PISA), which was first implemented by the Organization for Economic Cooperation and Development (OECD) in 2000, added a survey of students’ experiences of bullying in schools for the first time in 2015, including three types of bullying: relational bullying, verbal bullying, or physical bullying. In 2018, PISA continued to conduct a school-bullying survey in 75 countries and regions, showing the close concern that educators around the world have toward the problem. School bullying should be a high-priority concern for education policy makers and school administrators. Moreover, the PISA data includes those from both the students and the schools, which meets the requirements of multi-level analysis.

Therefore, based on the survey data of PISA 2018, our study used a multilevel analysis model and combines school-level and student-level variables to jointly explore various factors affecting school bullying and reveal the specific causes behind this phenomenon. Individual level variables included school bullying (including total school bullying, relational bullying, verbal bullying, and physical bullying), students’ gender, grade, education type, grade repetition, truancy and arriving late for class, family economic, social, and cultural status (ESCS), teacher support, and parent support they perceived, some of which were discussed above. School level variables included the describing characteristics of schools, such as school location, school type, school size, or school atmosphere, etc. The purpose of this paper is to investigate whether these factors have impacts on students’ bullying and what the effect of the impact is.

## 2. Materials and Methods

### 2.1. Materials

The data for this study came from the PISA 2018 survey database of four provinces and cities in mainland China (Beijing, Shanghai, Jiangsu, and Zhejiang). First, we downloaded the 2018 global Student questionnaire data file and School questionnaire data file from the PISA website https://www.oecd.org/pisa/data/2018database/ (accessed on 15 October 2021). For a brief introduction to PISA and descriptions of the questionnaires, see Appendix A. Then, we selected the data for mainland China. The student questionnaire data of mainland China includes 12,058 middle school students aged 15 (from 15 years and 3 months to 16 years and 2 months), and the school questionnaire data includes 361 schools. Finally, after deleting samples with missing data and those unable to meet the statistical criteria, the sample size of this study was 11,497 students from 334 schools (see Appendix A for detailed standards and procedures).

### 2.2. Research Variables

The variables of our study included individual-level variables of students and environment-level variables of school.

The student-level variables included the following: suffering from school bullying (including total school bullying, relational bullying, verbal bullying, and physical bullying), which was the outcome variable of the study; gender, grade, education type, grade repetition, truancy and arriving late for class, family economic, social, and cultural status (ESCS, teacher support, and parent support, which were predictor variables.

The school-level variables were divided into two categories. One was variables derived from the group level describing the characteristics of the schools, including school location, school type, school size, class size, student–teacher ratio, proportion of boys, proportion of special needs students, proportion of students without graduation certificates, student behaviors that hindered learning, and teacher behaviors that hindered learning. All of the above variables were completed by the principal (or principal representative) of each school. The other type was variables based on shared constructs, in which group characteristics were derived from combinations of group members and contained the attitudes, perceptions, or behaviors of group members. The variables were taken from student questionnaires, but they were aggregated by group, averaged, and then integrated into group variables, including school discipline atmosphere, competitive atmosphere among students, and cooperative atmosphere among students. See Appendix B for question descriptions, original corresponding items, and coding of these variables. Descriptive statistics of the above variables are shown in Table 1.

### 2.3. Statistical Analyses

The statistical software packages used in this study were SPSS 24 and HLM 6.08. The statistical methods included reliability and validity tests, a regression analysis, and a multilevel model analysis.

In many social science research fields, such as psychology, education, and management, data are often in nested structures (nested data, multilayer data, or multilevel data) where, for example, students are nested under a class, and the class is under the school. The sample data from such nested structures are generally not independent, which violates traditional statistical assumptions (residual independence and a homogeneous regression slope). Using a traditional OLS regression method to analyze nested samples and, thus, ignoring the problem of hierarchical differences can bring about statistical estimation bias. If the conclusions obtained from a high-level data analysis are inferred with lower-level data, it is easy to overestimate the lower-level conclusions, resulting in “ecological fallacy”. Conversely, if the conclusions obtained from a lower-level data analysis are inferred with higher-level data, this leads to “atomistic fallacy” [11]. Therefore, it is very important to understand the variation caused by different groups using a multilevel analysis method.

An analysis for nested data has been gradually developed, and finally in the 1990s, a complete and systematic theory and method was developed, namely the multilevel model analysis or multilevel analysis (HLM); in addition, HLM software was designed for multilevel analyses. Using the multilevel analysis method enables the analysis of multilevel data in one model at the same time, reducing the statistical errors mentioned above, and it can analyze the possible interactions between different levels’ data, describing the characteristics of a phenomenon more objectively. In this study, using HLM software, we combined school-level variables and student-level variables to explore the influencing factors that affected school bullying and attempted to reveal the specific causes behind this phenomenon.

A structure diagram of this study is shown in Figure 1.

## 3. Results

As suggested by Bryk and Raudenbush [12], a multilevel analysis should include the implementation of four sub-models: Null Model, Random Coefficient Model, Intercepts as Outcomes Model, and Slopes as Outcomes Model. Since our study did not specifically explore the moderating effects of the school-level contextual variable group, Slopes as Outcomes Model was not performed. Therefore, this study analyzed three multilevel models (see Table 2 for total school bullying): Model I (Null Model) was used to test the proportion of group variation to the overall variance in student suffering from school bullying and to three different types of bullying (that is, the contextual effect between different schools), which provided a reasonable basis for a subsequent multilevel analysis to confirm the intraclass correlation coefficient (ICC) of the dependent variable, and the between-group variation component could meet the requirements for performing a multilevel model analysis. Model II (Random Coefficient Model) was used to test the direct impacts of student-level variables on school bullying. Model III (Intercepts as Outcomes Model) was used for testing the direct impacts of school-level variables on school bullying. Model III was the full model for this study.

### 3.1. Model I: Null Model

No explanatory variables were included in the Null Model; instead, it only contained the result variables, and the corresponding formula is shown in Appendix C.

As shown in the results of Table 2, the between-group variation component (τ_00_) of suffering from school bullying was significantly different from 0 (χ^2^ = 444.065, *p* < 0.001), indicating that the degree of student suffering from school bullying in the same school was similar, but there were significant differences in different schools. Similarly, as shown in Table A1, Table A2 and Table A3 (these three tables can be seen in Appendix C), the between-group variance components (τ_00_) of relational bullying (χ^2^ = 409.931, *p* = 0.003), verbal bullying (χ^2^ = 387.921, *p* = 0.020), and physical bullying (χ^2^ = 470.758, *p* < 0.001) were also significantly different from 0. These results illustrate that the variation between groups could not be ignored. In order to avoid biased interpretation of the results, it was necessary to use the multilevel model for data analysis.

### 3.2. Model II: Random Coefficient Model

In this model, the student-level variables are not uncentered, with the exception of grades, teacher support, and parent support, which were generally grand-centered. Kreft pointed out that categorical variables must not be mean centered [13]. If a continuous variable is meaningful for 0, it does not need to be centered because, whether it is centered or not, it has no effect on the estimated value and significance but only affects the interpretation of the results. To make an interpretation meaningful, it needed to be mean-centered in both Level 1 (student-level) and Level 2 (school-level) and always use grand-centered variables which is equivalent to the original data, while group-centered variables are not equivalent to the original data [13]. The corresponding formula is shown in Appendix C.

From the results in Table 2, it can be seen that, in addition to grade and education type, gender, grade repetition, and truancy and arriving late for class at the individual level all had significant positive explanatory powers on students’ total school bullying. Both teacher support and parent support have significant negative explanatory powers on students’ total school bullying.

Similarly, relational bullying, verbal bullying, and physical bullying showed the same effects, as shown in Table A1, Table A2 and Table A3 (these three tables can be seen in Appendix C). When teacher support increased by 1, total school bullying decreased by 0.562, relational bullying decreased by 0.195, verbal bullying decreased by 0.180, and physical bullying decreased by 0.188. When parent support increased by 1, total school bullying decreased by 0.391, relational bullying decreased by 0.142, verbal bullying decreased by 0.133, and physical bullying decreased by 0.117. Lastly, ESCS only had a significant negative explanatory power on students suffering from total school bullying and physical bullying but had no significant effect on relational bullying and verbal bullying. When family ESCS increased by 1, total school bullying decreased by 0.066, and physical bullying decreased by 0.042.

The above results indicate that boys suffered from a greater degree of school bullying than girls (including total school bullying and three types of bullying), and students who repeated grades, were truant, and arrived late in the past two weeks were more severely bullied than those who had not. The lower the student family ESCS is, the higher the levels of school bullying and physical bullying are. The lower students perceived teacher support and parent support, the more severe the school bullying.

### 3.3. Model III: Intercepts as Outcomes Model

In this model, with the exception of the variables of school location, school type, proportion of special needs students, and proportion of students without graduation certificates being uncentered, all the other variables were grand-centered.

The results in Table 2 and Table A1, Table A2 and Table A3 (these three tables can be seen in Appendix C) show that only the two variables of school discipline atmosphere and competitive atmosphere among students had a significant impact on student suffering from total school bullying and three other types of bullying. The variable of school discipline atmosphere had a significant negative explanatory power on the degree of student suffering from bullying, while the variable of competitive atmosphere among students had a significant positive explanatory power on the degree of student suffering from bullying. The analysis results of Model III show that, when the school discipline atmosphere increased by 1, school bullying decreased by 0.572, relational bullying decreased by 0.198, verbal bullying decreased by 0.143, and physical bullying decreased by 0.231. However, when the variable of competitive atmosphere among students increased by 1, school bullying increased by 0.806, relational bullying increased by 0.263, verbal bullying increased by 0.289, and physical bullying increased by 0.245.

These results indicate that the worse a school’s discipline atmosphere is, the more severe the level of school bullying students experienced is, i.e., students in schools with poor discipline atmospheres experienced a greater degree of school bullying than those in schools with better discipline atmospheres. However, the higher the competitive atmosphere among students is, the higher the level of school bullying students experienced is, i.e., students in schools with high inter-student competition atmospheres experienced greater levels of bullying than those in schools with low inter-student competition atmospheres.

The remaining variables of school background characteristics all did not have significant impacts on students’ suffering from school bullying or the three other types of bullying.

## 4. Discussion

### 4.1. Influence of Student-Level Variables on Students Being Bullied

According to the results in Model II, in addition to grade and education type, the student-level variables of gender, grade repetition, and truancy and arriving late for class all have significant positive effects on total school bullying and the three types of bullying, while teacher support and parents’ support both have significant negative explanatory power on students’ total school bullying and the three types of bullying. ESCS only negatively affects students’ total school bullying and physical bullying but not relational bullying and verbal bullying. The results above show that boys suffer from a greater degree of school bullying than girls, and students who have repeated grades, who are truant, and who have been late for class in the past two weeks are more severely bullied than those who have not. The lower the family’s ESCS is, the higher the level of total school bullying and physical bullying are. The lower the perceived teacher support and parents support are, the more severe the school bullying is. The above results are discussed further below.

First, in our study, we found that boys suffered more severe bullying than girls, both for total school bullying and for the three different types of bullying, which is partly consistent with previous studies. Previous studies have confirmed that boys are at greater risk of school bullying than girls [8,14]. In terms of different types of bullying, previous studies have found that girls are more susceptible to relational bullying [15], and data from OECD countries also show that girls are more likely to be exposed to “spreading rumors by other students” [16], while boys are more likely to suffer from physical bullying [17]. Based on the data analysis of PISA 2015, Huang found that boys were more prone to physical bullying than girls [5], such as physical hitting or pushing. In addition, boys were also more likely to experience verbal bullying than girls, such as being teased by others. In summary, boys are at greater risk of bullying than girls. The reason for this may be that boys are more prone to agitation and conflict than girls, which makes boys significantly more likely than girls to be bullied or to bully others. Therefore, we should pay more attention to the male group and give them more help related to the phenomenon of bullying.

Second, our study found that students who repeated grades, had absenteeism, and were late to class within the last two weeks were more likely to be bullied at school. In addition to poor academic performance, repeat-grade students may also have difficulties with the development of social and emotional skills. When these older students study and live with new, younger classmates, they may be very easily discriminated against, laughed at, or teased by other students and may even be socially excluded [18], which may, in turn, lead to bullying incidents [19]; this is similar for students with absenteeism and lateness. According to previous studies, disciplinary violations such as those for truancy, skipping class, and lateness may be external manifestations of students rejecting learning. If students are unwilling to enter a classroom, or even skip class, it is naturally difficult for them to achieve good academic performances [20], while students with poor academic performances are more likely to be bullied, which has been confirmed by previous research [10,21]. On the other hand, those who are bullied protect themselves by avoiding school or being truant, and then these truant students have more difficulties keeping up with teaching or are unable to obtain help from the school due to not showing up to school on time. This also weakens the connection between students and the school environment, leading to poor academic achievement [22,23]. Therefore, there may be a mutual causal relationship between truancy, absenteeism, lateness, and other disciplinary violations and students being bullied on campus. To reduce bullying on campus, educators can start with the strict management of students’ disciplinary violations to ensure that students can attend school on time because this is a premise to ensure quality of learning. In this way, it is possible to improve their academic achievements and help them establish a close relationship with the school, making it easier to seek help from teachers and classmates, which is conducive to reducing the occurrence of bullying.

Furthermore, the results show that students’ ESCS had a significant negative impact on students’ total school bullying and physical bullying, which meant that the lower a student’s family economic sociocultural status is, the higher the degrees of overall school and physical bullying is. This result is consistent with the results of Huang and Zhao [23], as well as empirical research from the Netherlands, which showed that adolescents with lower social status had a higher proportion of physical and psychological symptoms, which were more likely to be aggression by peers [24]. Therefore, schools and teachers should pay more attention to students from disadvantaged backgrounds and carry out targeted psychological counseling and assistance to reduce the risk of bullying for these students.

Finally, this study found that the higher the levels of teacher support and parent support perceived by students are, the lower the level of suffered school bullying is. A close parent–child relationship can help students obtain more help when they suffer from school bullying, and parents who care about and support their children emotionally can not only help their children decrease school bullying but can also relieve children’s psychological pressure and pain after students are bullied [25]. In terms of teacher–student relationships, teachers’ actions of supporting, caring about students’ academic progress, and expecting students’ success make students feel accepted, respected, and cared for. On the one hand, students can better seek help from teachers; on the other hand, closeness and harmony of teacher–student relationships greatly reduces the chance of negative interpersonal behaviors, such as bullying [26]. Thus, both families and schools are key forces to fight against school bullying, and home–school cooperation can better build antibullying barriers in students’ lives and learning.

In conclusion, when developing school bullying interventions, more attention should be paid to male students, students who repeat grades, are late, or are absent from class, and students with lower ESCS. For example, physical bullying of male students should be paid concern. Pushing, beating, and other similar behaviors should be stopped in time. As for students with low academic performance, parents should encourage and support their children rather than criticize and blame them. Teachers should also pay more attention to students who are often late, absent, or from lower backgrounds and should strengthen their ability to recognize bullying incidents, especially the two types of relationship bullying and verbal bullying, because they will not cause obvious physical harm, making it very difficult to identify. In addition, teachers can pay close attention to the way students make friends and interact with each other. They can observe whether a particular student is excluded or isolated in group activities, PE class, and after class. Once they find signs of bullying, appropriate treatment should be provided the first time to prevent the occurrence of the event. Finally, teachers should consult more professional counselors, attend seminars on school bullying cases, and flexibly use effective ways to deal with bullying cases to reduce the harm caused by bullying.

### 4.2. Influence of School-Level Variables on Students Being Bullied

The results of Model III show that only the school discipline atmosphere and the competitive atmosphere among students of the school environment level variables have significant impacts on the total school bullying and the three types of bullying. School discipline atmosphere has a significant negative explanatory power on school bullying, while the competitive atmosphere among students has a significant positive explanatory power on students being bullied, indicating that students in schools with a worse discipline atmosphere experience greater levels of bullying than students in schools with a better one; students in schools with a high level of inter-student competition are more likely to experience higher levels of school bullying than those in schools with a lower level. A good school discipline atmosphere helps protect students and make them less vulnerable to school bullying [23], but the competitive atmosphere among students may make some students feel jealous or hate other classmates, which in turn increases the chances of students being bullied at school.

First, the negative correlation between school disciplinary atmosphere and students suffering from bullying has been confirmed by some studies [2,18,27]. A good school discipline atmosphere helps to protect students and make them less vulnerable to school bullying [23]. The reason for this may be because when students learn and interact in a well-ordered environment, they are often more willing to engage in it because they feel safe, and the trigger factors for student aggressive behavior are greatly reduced [28]. Therefore, when formulating plans to prevent school bullying at the school level, more consideration should be given to the important role of school disciplinary atmosphere, which is not only an invisible school culture but also can be reflected in the implementation of school rules and discipline. In addition, this is also consistent with strengthening the management of skipping class, truancy, lateness, and other disciplinary violations mentioned above.

Second, atmospheres of competition and cooperation among students in schools are important aspects of the school climate [29]. This is the first time that this topic has appeared in the PISA questionnaire survey. The results show that competitive atmosphere had a significant positive explanation for the degree of school bullying. However, cooperative atmosphere did not have a significant impact on student suffering from school bullying. This is an interesting result, suggesting that competitive and cooperative atmospheres at the school level may not be two opposing aspects, and they may have their own working principles.

The positive association between perceived competitive atmosphere and school bullying has been supported by some studies. Volk proposed that a competitive atmosphere may make some students feel jealous or hateful toward other classmates, which in turn increases the chances of students being bullied at school [30]. Wang’s research showed that both academic competition and social competition perceived by primary and secondary school students were significantly positively correlated with school bullying [31]. In another research project, Wang proposed that a vicious, competitive atmosphere formed among students not only led to campus bullying [32] but also generated countless indifferent bystanders who saw the campus bullying but were unwilling to lend a helping hand. Therefore, when intervening in bullying in schools, students should be consciously guided to engage in positive and benign competition and to avoid vicious competition. In this way, good peer relationships in the school environment form, and the occurrence of bullying is reduced.

Finally, a surprising result in our study is that student behaviors and teacher behaviors that hinder learning in schools both had no effect on students’ experiences of school bullying. A previous study regarded these two variables as a measure of school spirit [33]. Through a multilevel analysis based on PISA 2015 data from four provinces and cities in China, the study found that student and teacher behaviors that hindered learning had significant negative impacts on students’ scientific literacy without controlling for student and school ESCS values; however, when controlling for them, the effects were smaller and no longer significant. Due to the large number of control variables involved in our study and different combinations of control variables producing various different results, this part may therefore need to be further explored in future studies.

Here are some suggestions on the results. Students spend a lot of time in school. As an important place of education, school plays a decisive role in the formation of students’ personality and behavior. When the school atmosphere is positive and friendly, bullying can be reduced. Schools should instruct students to learn ways to protect themselves, identify bullying in schools, and seek help from teachers and classmates to better protect themselves. Schools should strengthen the moral education of students and cultivate students’ good sense of justice and moral sense, making students brave enough to stop school bullying or report bullying to teachers.

The psychological counseling institution of schools should play an active role in school bullying and treat every bullying case as a major campus crisis. In addition to isolation, placement, and counseling, it is important to continuously observe and follow up the development of physical and mental status of the cases, both of the perpetrator and the victim, making sure they are physically and mentally healthy. In addition, schools should strengthen students’ interpersonal communication and life education. Students must learn to respect each other’s lives and cherish their own. They must understand that certain behaviors should not be allowed, such as making fun of each other’s sexual orientation and physical characteristics. It must be made clear that students’ bullying behavior may directly or indirectly kill their classmates. When students are aware that bullying can have such serious consequences, it may be effective in reducing the incidence of school bullying.

At last, bullying should not be seen as a problem of a few but as one of society’s problems. The whole society should work together to create a friendly campus environment. Education authorities should integrate schools, neighboring communities, police and government organs, social welfare organizations, mental health units and other relevant social resources, and professional assistance to provide students with the most appropriate treatment. In order to prevent bullying, teachers and schools are encouraged to make more use of social resources. A variety of professional teams, including school principals, directors, tutors, psychological consultants, students, parents, juvenile police officers, social workers of welfare organizations, and other experts should work together to investigate, evaluate, and formulate counselling programs to effectively reduce bullying incidents in schools.

### 4.3. Limitations and Future Research Directions

Because the selected variables were all obtained from the PISA test and were limited by the scope of the database, our study may not contain all the factors that affect students’ experiences of school bullying. In follow-up research, other methods, such as using other databases or questionnaires made by the researchers themselves, should be used to include more influence factors to analyze this topic. In addition, some factors in our study were not obviously related to school bullying, such as grade and education type of student-level, student behaviors and teacher behaviors that hindered learning, and cooperative atmosphere among students of school-level, so more rigorous field investigations may be needed [34].

Furthermore, regarding the multilevel analysis model, the analysis method of a multi-level mediation model can also be considered to further elaborate the specific operation paths of these influencing factors [35]. A multilevel mediation model can set the possible influence paths for factors that were found to have significant explanatory power based on the existing research so that all the independent variables can be included in the model at one time. It can provide more specific reference information for educational administrators and can broaden the scope of related research topics, providing more theoretical significance.

## 5. Conclusions

Based on the PISA 2018 survey data (including student data and school data) and using a multilevel analysis model, this study explored the impacting factors that affected school bullying. While the results are consistent with some previous studies, there are some new developments.

In the student-level variables, boys, students who have repeated grades, who are truant, and who have been late for the class in the past two weeks, and students whose economic, social, and cultural status is lower suffer from more severe school bullying. Furthermore, students who have perceived lower teacher support and parents support are more severely bullied than those who have not. Thus, when developing school bullying interventions, more attention should be paid to these students. Teachers and parents should give more emotional support to them. Additionally, school administrators, consulting teachers, and other relevant personnel should pay more attention to students. Once they find signs of bullying, appropriate treatment should be provided the first time to prevent the occurrence of the event.

In the school-level variables, students in schools with a worse discipline atmosphere and a higher level of inter-student competitive atmosphere experience greater levels of bullying. Therefore, when formulating plans to prevent school bullying at the school level, more consideration should be given to the important role of strengthening school disciplinary atmosphere and reducing the competitive atmosphere. A safe, well-ordered, positive, and friendly school environment helps protect students and make them less vulnerable to school bullying, and the establishment of such a campus environment needs the efforts of the whole society.

## Figures and Tables

**Figure 1 children-10-00653-f001:**
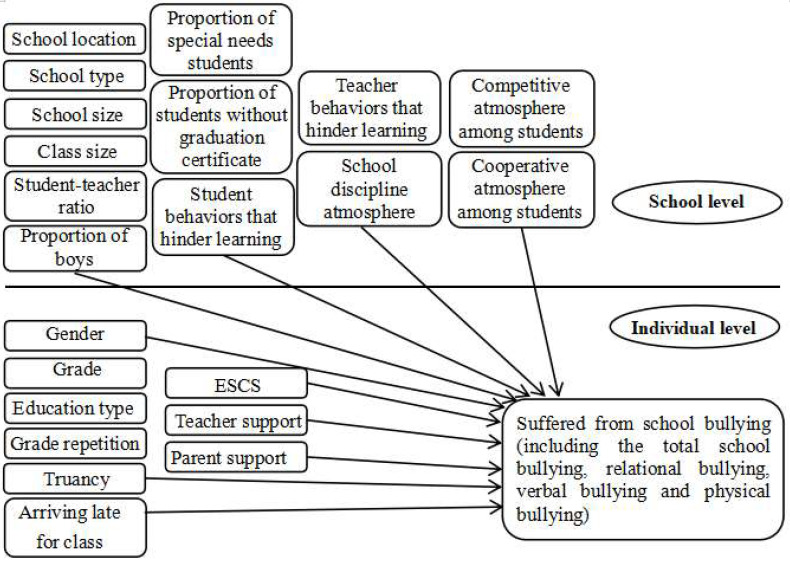
The structure of the study.

**Table 1 children-10-00653-t001:** Descriptive statistics of the variables.

Variable Name	Minimum	Maximum	Average	S.E.
Student-level variables				
School bullying	6.000	24.000	7.610	2.776
Relational bullying	2.000	8.000	2.494	1.077
Verbal bullying	2.000	8.000	2.542	1.038
Physical bullying	2.000	8.000	2.579	1.051
Gender (female, male)	0	1.000	0.521 ^1^	0.500
Grade	7.000	12.000	9.640	0.549
Education type (general education, vocational education)	0	1.000	0.181	0.385
Grade repetition (no, yes)	0	1.000	0.063 ^1^	0.242
Truancy (no, yes)	0	1.000	0.075	0.264
Arriving late for class (no, yes)	0	1.000	0.302	0.459
Economic, social, and cultural status (ESCS)	−5.077	3.102	−0.359	1.089
Teacher support	1.000	4.000	3.393	0.693
Parent support	1.000	4.000	3.330	0.643
School-level variables				
School location (town schools, city schools)	0	1.000	0.630	0.485
School type (public school, private school)	0	1.000	0.140 ^1^	0.345
School size	78.000	13,400.000	1926.920	1461.488
Class size	18.000	53.000	38.760	8.003
Student–teacher ratio	1.000	100.000	10.755	6.261
Proportion of boys	0.100	0.890	0.522	0.084
Proportion of special needs students	0	107.000	7.686	11.094
Proportion of students without graduation certificates	0	42	1.230	3.709
Student behaviors that hinder learning	5.000	20.000	11.332	5.085
Teacher behaviors that hinder learning	5.000	20.000	11.320	4.429
School discipline atmosphere	2.900	3.830	3.411	0.166
Competitive atmosphere among students	2.240	3.010	2.566	0.146
Cooperative atmosphere among students	2.330	3.520	2.848	0.176

^1^ For categorical variables, the Average refers to the percentage of the latter category in each variable. For example, for Gender, 0.521 represents that the proportion of male students is 52.1%; for Grade repetition, 0.063 represents that the proportion of students’ repeating grade is 6.3%; and for School type, 0.140 represents that the proportion of private schools is 14%. Other categorical variables are similarly interpreted.

**Table 2 children-10-00653-t002:** Multilevel analysis results of the influencing factors of students suffering from school bullying.

	Model I			Model II			Model III		
Fixed Effect	γ Coefficient	*S.E.*	*p*	γ Coefficient	*S.E.*	*p*	γ Coefficient	*S.E.*	*p*
γ_00_	7.610	0.030	<0.001	6.928	0.035	<0.001	6.962	0.064	<0.001
Student-level variables									
Gender γ_10_				0.851	0.052	<0.001	0.842	0.050	<0.001
Grade γ_20_				−0.071	0.035	0.180	−0.089	0.054	0.099
Education type γ_30_				0.028	0.080	0.730	0.033	0.077	0.667
Grade repetition γ_40_				0.390	0.127	0.003	0.387	0.111	0.001
Truancy γ_50_				1.151	0.128	<0.001	1.138	0.097	<0.001
Arriving late for class γ_60_				0.331	0.063	<0.001	0.324	0.056	<0.001
ESCS γ_70_				−0.066	0.027	0.015	−0.056	0.027	0.041
Teacher support γ_80_				−0.562	0.044	<0.001	−0.554	0.044	<0.001
Parent support γ_90_				−0.391	0.043	<0.001	−0.388	0.044	<0.001
School-level variables									
School location γ_01_							−0.008	0.060	0.891
School type γ_02_							−0.051	0.086	0.552
School size γ_03_							<0.001	<0.001	0.813
Class size γ_04_							0.002	0.004	0.582
Student–teacher ratio γ_05_							−0.001	0.006	0.901
Proportion of boys γ_06_							−0.178	0.337	0.598
Proportion of special needs students γ_07_							<0.001	0.003	0.980
Proportion of students without graduation certificates γ_08_							−0.007	0.008	0.355
Student behaviors that hinder learning γ_09_							<0.001	0.011	0.975
Teacher behaviors that hinder learning γ_10_							0.002	0.013	0.865
School discipline atmosphere γ_011_							−0.572	0.192	0.004
Competitive atmosphere among students γ_012_							0.806	0.201	<0.001
Cooperative atmosphere among students γ_013_							−0.218	0.177	0.220
Random effects	Variance components	χ^2^	*p*	Variance components	χ^2^	*p*	Variance components	χ^2^	*p*
τ_00_	0.071	444.065	<0.001	0.051	363.704	0.010	0.047	339.720	0.023
σ^2^	7.632			6.739			6.729		

## Data Availability

The original data of the study can be found in the website https://www.oecd.org/pisa/data/2018database/ (accessed on 15 October 2021).

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
