# Peer review of "A Multilevel Analysis of Factors Influencing School Bullying in 15-Year-Old Students"

_children, 2023, doi:10.3390/children10040653_

Round 1

Reviewer 1 Report

Thank you for conducting this study (analysis). It is a well-written manuscript. I have few notes that may improve the manuscript better. 

1. Abstract: using abbreviation (PISA and ESCD) may confuse the reader before reading the full article. Also abstract could be better if you add any recommendation statement(s) at the end of the abstract. 

2. Introduction: addressed main study variables and aims. 

3. Methods: 

a) It will be better if the authors have a description about the procedure of data collection (sampling methods). 

b) I do not think there is a need for table 1. It might be in appendix if you wish. You may delete this table and replace the variable description (the third column) with ranges in table 2. (in other words, merge table 1 and 2 into one table). In the same issue, I ques it is better to have tables A1, A2, and A3 in the text (not in appendix). 

c) statistical analysis plan is clear and figure 1 is excellent too. 

4. Results

a) I do not think there are need for including the  equations (B, Y, U<<<). It does not add anything to the reader. I think to have only your description of the results without the equations. 

5. Disscussion: excellent, you explain all the significant results

6. I think you may strengthen your study by adding a paragraph that contains recommendations for researchers, teachers, parents, and may be to mental healthcare providers who are working in the schools.

Finally, be careful about using the concept of environmental-level and school-level. You used them interchangabley.  

Thank you 

Author Response

Responses to Reviewer 1 Comments

Thank you for conducting this study (analysis). It is a well-written manuscript. I have few notes that may improve the manuscript better.

Authors’ Response: Thank you for your affirmation for our manuscript and suggestions. We appreciate your positive comments and have worked to address your concerns. We value the time and effort you spent in giving constructive revisions.

Point 1: Abstract: using abbreviation (PISA and ESCD) may confuse the reader before reading the full article. Also abstract could be better if you add any recommendation statement(s) at the end of the abstract.

Response 1: Thanks for your suggestion. We have added the full names to the abbreviations (PISA and ESCS). Also, we added recommendation statements at the end of the abstract, and revised in the abstract.

Line 10-36

Abstract: Background: School bullying causes serious impacts on adolescents’ physical and mental health few studies have explored the various factors influencing bullying by combining different levels of data. Methods: Based on the database of four Chinese provinces and cities of the Program for International Student Assessment (PISA) in 2018 year, this study used a multilevel analysis model that combined school-level variables and student-level variables to explore the influencing factors of students' being bullied. Results: Students’ gender, grade repetition, truancy, and arriving late for class, economic, social, and cultural status (ESCS), teacher support and parent support had significant explanatory power for school bullying on the student-level; on the school-level, school discipline atmosphere and competitive atmosphere among students had significant impacts on school bullying. Conclusions: Boys, students who have repeated grades, truancy and are late for the class, and students with lower ESCS suffer from more severe school bullying. When developing school bullying interventions, teachers and parents should pay more attention to those students, and provide more emotional support and encouragement to them. Meanwhile, students in schools with a low-er discipline atmosphere and a higher level of competitive atmosphere experience greater levels of bullying, and schools should create more positive and friendly environment to prevent bullying events.

Point 2: Introduction: addressed main study variables and aims.

Response 2: We have made a breif introduction about main study variables and aims in the last paragraph of the Introduction part.

Line 129-139

Therefore, based on the survey data of PISA 2018, our study used a multilevel analysis model and combines school-level and student-level variables to jointly explore various factors affecting school bullying and reveal the specific causes behind this phenomenon. Individual level variables included school bullying (including total school bullying, relational bullying, verbal bullying, and physical bullying), student’s gender, grade, education type, grade repetition, truancy, and arriving late for class; family economic, social, and cultural status (ESCS); and teacher support and parent support they perceived, some of which were discussed above. School level variables included the describing characteristics of schools, such as school location, school type, school size, or school atmosphere, etc. The purpose of this paper is to investigate whether these factors have impacts on students' bullying, and what is the effect of the impact.

Point 3: a) It will be better if the authors have a description about the procedure of data collection (sampling methods).

Response 3: a) Thanks for your advice. We have added more detailed information about sampling methods, see in Appendix A for 2. The procedure of data collection.

Line 724-748

  1. b) I do not think there is a need for table 1. It might be in appendix if you wish. You may delete this table and replace the variable description (the third column) with ranges in table 2. (in other words, merge table 1 and 2 into one table). In the same issue, I ques it is better to have tables A1, A2, and A3 in the text (not in appendix).

Response 3: b) Thank you very much for your suggestion. There was indeed some duplication in the previous tables, and we have merged table 1 and 2 into one table. However, to keep the main text concise, tables A1, A2, and A3 remains in appendix, because the content in these forms is the same as Table 2, only for different types of bullying.

Line 188-190

  1. c) statistical analysis plan is clear and figure 1 is excellent too.

Response 3: c) Thank you very much for your compliment.

Point 4: I do not think there are need for including the equations (B, Y, U<<<). It does not add anything to the reader. I think to have only your description of the results without the equations.

Response 4: Thanks for your suggestion. We have already deleted the equations in the main text, but put this content in Appendix C, in case some researchers may want to refer to these information, which provide background understanding on the Greek letters in the Table 2 and Tables A1, A2, and A3.

Line 1027-1079

Point 5: Disscussion: excellent, you explain all the significant results.

Response 5: Thanks again for your compliment.

Point 6: I think you may strengthen your study by adding a paragraph that contains recommendations for researchers, teachers, parents, and may be to mental healthcare providers who are working in the schools.

Finally, be careful about using the concept of environmental-level and school-level. You used them interchangabley.

Response 6: Thanks for your useful recommendation. We have added paragraphs that contains recommendations both in individual-level of teachers, parents, and mental healthcare providers and in envirionment-level of school administrator, police and government organs, social welfare organizations, mental health units and other relevant social resources and professional assistance.

Line 504-520

In conclusion, when developing school bullying interventions, more attention should be paid to male students, students who repeat grades, are late, or are absent from class, and students with lower ESCS. For example, physical bullying of male students should be paid be concern. Pushing, beating, and other similar behaviors should be stopped in time. As for students with low academic performance, parents should encourage and support their children rather than criticize and blame them. Teachers should also pay more atten-tion to students who are often late, absent or from lower backgrounds, and should strengthen their ability to recognize bullying incidents. Especially the two types of rela-tionship bullying and verbal bullying, because they will not cause obvious physical harm, making it very difficult to identify. In addition, teachers can pay close attention to the way students make friends and interact with each other. They can observe whether a particular student is excluded or isolated in group activities, PE class and after class. Once they find signs of bullying, appropriate treatment should be provided in the first time to prevent the occurrence of the event. Finally, teachers should consult more professional counselors, at-tend seminars on school bullying cases, and flexibly use effective ways to deal with bully-ing cases to reduce the harm caused by bullying.

Line 586-616

Here are some suggestions on the results. Students spend a lot of time in school. As an important place of education, school plays a decisive role in the formation of students' personality and behavior. When the school atmosphere is positive and friendly, bullying can be reduced. Schools should instruct students to learn ways to protect themselves, identify bullying in schools, and seek help from teachers and classmates to better protect themselves. Schools should strengthen the moral education of students, cultivate students' good sense of justice and moral sense, making student to be brave enough to stop school bullying, or report bullying to teachers.

The psychological counseling institution of schools should play an active role in school bullying and treat every bullying case as a major campus crisis. In addition to iso-lation, placement and counseling, it is important to continuously observe and follow up the development of physical and mental status of the cases, both the perpetrator and the victim, making sure they are physically and mentally healthy. In addition, schools should strengthen students' interpersonal communication and life education. Students must learn to respect each other's lives and cherish their own. They must understand that cer-tain behaviors should not be allowed, such as making fun of each other's sexual orienta-tion and physical characteristics. It must be made clear that students' bullying behavior may directly or indirectly kill their classmates. When students are aware of that bullying can have such serious consequences, it may be effective in reducing the incidence of school bullying.

At last, bullying should not be seen as a problem of a few, but as one of society's prob-lems. The whole society should work together to create a friendly campus environment. Education authorities should integrate schools, neighboring communities, police and government organs, social welfare organizations, mental health units and other relevant social resources and professional assistance to provide students with the most appropri-ate treatment. In order to prevent bullying, teachers and schools are encouraged to make more use of social resources. A variety of professional teams, including school principals, directors, tutors, psychological consultant, students, parents, juvenile police officers, social workers of welfare organizations and other experts, should work together to investigate, evaluate and formulate counselling programs to effectively reduce bullying incidents in schools.

Reviewer 2 Report

Bullying in schools is a prevalent problem. It occurs all around the world, with a higher prevalence in some areas and among certain genders. The current study aims to look into the concerns among 15-year-old middle school students. Why is this age group mentioned in the title?

There is no conclusion or advice in the abstract. It is advised to leave out the numbering in each subsection.

Because many studies on school bullying had been published, the introduction needed to indicate a clear study gap. Part of the historical information in the introduction can be omitted or summarised.

It is preferable if the authors can discuss the study design, study area, source and reference population, sample size determination, and sampling procedure. It is assumed that the authors used data from the PISA 2018 student questionnaire, hence a description of the questionnaire is provided.

The description of the components in Table 2 is unclear; what does average mean (particularly when dealing with categorical data)?

The conclusion and discussion were well written.

Author Response

Responses to Reviewer 2 Comments

Point 1: Bullying in schools is a prevalent problem. It occurs all around the world, with a higher prevalence in some areas and among certain genders. The current study aims to look into the concerns among 15-year-old middle school students. Why is this age group mentioned in the title?

Response 1: Thanks for your comments. The reason this age group mentioned in the title is that the data materials used in our study are from the 2018 PISA test, and the subjects are all 15 years old middle school students.

Point 2: There is no conclusion or advice in the abstract. It is advised to leave out the numbering in each subsection.

Response 2: We have added conclusion including recommendation advice at the end of the abstract, and deleted the numbering in each subsection.

Line 10-36

…Conclusions: Boys, students who have repeated grades, truancy and are late for the class, and students with lower ESCS suffer from more severe school bullying. When developing school bullying interventions, teachers and parents should pay more attention to those students, and provide more emotional support and encouragement to them. Meanwhile, students in schools with a lower discipline atmosphere and a higher level of competitive atmosphere experience greater levels of bullying, and schools should create more positive and friendly environment to prevent bullying events.

Point 3: Because many studies on school bullying had been published, the introduction needed to indicate a clear study gap. Part of the historical information in the introduction can be omitted or summarised.

Response 3: We have added some expressions to indicate the study gap. Also, we have deleted some historical information, such as the prevalence of bullying surveyed by other international institutions, leaving only the PISA data.

Line 112-119

In conclusion, it can be seen that school bullying is affected by various factors of individuals, families, and schools, but there have been some contradictions among past studies, such as age and school type, which may be related to sampling or research methods. In addition, few studies have explored the various factors influencing bullying by combining different levels of data…

Line 50-74

Point 4: It is preferable if the authors can discuss the study design, study area, source and reference population, sample size determination, and sampling procedure. It is assumed that the authors used data from the PISA 2018 student questionnaire, hence a description of the questionnaire is provided.

Response 4: Thanks for your kind suggestions. We have added more detailed information about descriptions of student and school questionnaires, and sampling methods, see in Appendix A.

Line 691-748

Appendix A

  1. The introduction to PISA and descriptions of student and school questionnaires
  2. The procedure of data collection

Point 5: The description of the components in Table 2 is unclear; what does average mean (particularly when dealing with categorical data)?

Response 5: Thanks for your question. In the process of modification, we combined Table 1 and 2, and explained the meaning of category variables in the annotation of Table 1. And we converted all the category variables in the study into dummy variable.

Line 192-195

For categorical variables, the Average refers to the percentage of the latter category in each variable. For example, for gender, 0.521 represents the proportion of male students is 52.1%; for Grade repetition, 0.063 represents the proportion of students’ repeating grade is 6.3%; and for School type, 0.140 represents the proportion of private schools is 14%. Other categorical variables are similarly interpreted.

Point 6: The conclusion and discussion were well written.

Response 6: Thank you for your affirmation.

Reviewer 3 Report

Thank you for the opportunity to review this paper which examines individual and contextual factors influencing school bullying in a large sample 11497 of 15 year old students in China. The manuscript is well written and methods are appropriate. However, I suggest authors to address the issued listed below before the paper be considered for publication.

Title: I suggest authors to review the title. It is a multilevel analysis (not study) of factors influencing school bulling in 15-year-old students (given that the sample does not focus on students suffering from bullying exclusively.)

Abstract

Please refrain from using acronyms like PISA and ESCS in the abstract.

Introduction

The first entry line 28-30 is redundant, it can be cut.

 Methods

I understand part of the data (individual variables) were obtained through self-report questionnaires filled out by students, whereas other school related variables were obtained by data provided by school principles. This should be clarified in the Materials section, in order to avoid potential confusion.

Results

All the descriptive results should be presented succinctly in the Table 2 where min - max range should be added for continuous variables in order for mean levels to be read correctly and N, /% should be provided for categorical variables so that significant differences in proportions can be easily read. Reporting results correctly and comprehensively in Table 2 shall substitute all the text provided in the  section 3.1 which is at this point redundant as it duplicates results already reported in the table.

Discussion

Authors should start the discussion section 4.1 with their own findings rather than with findings of previous research. I suggest they move up lines 370 - 374 in the beginning and discuss how their results fit in or not with existing research.

The same applies to section 4.2 of the discussion.

Line 473-474 calls for control analysis to be performed by authors. I suggest they either review this paragraph or perform additional analysis to test their interpretation of findings.

Limitations

It is not clear what the authors intend by “self-made questionnaires”? Please clarify. Also they state that “some factors were not related to school bullying” – please specify, which factors?

Conclusions

The conclusion section should not be a dry repetition of the study results but it should contain take-home points based on the results and the potential impact that findings may have for research and practice. I suggest authors to review their conclusions from line 494 to line 512.

Author Response

Thank you for the opportunity to review this paper which examines individual and contextual factors influencing school bullying in a large sample 11497 of 15 year old students in China. The manuscript is well written and methods are appropriate. However, I suggest authors to address the issued listed below before the paper be considered for publication.

Authors’ Response: We appreciate your affirmation and positive comments on our manuscript. We have worked to revise it according to you suggestion and hope the revised version could be acceptable for you.

Point 1: I suggest authors to review the title. It is a multilevel analysis (not study) of factors influencing school bulling in 15-year-old students (given that the sample does not focus on students suffering from bullying exclusively.)

Response 1: Thanks for your suggestion. We have modified the title from “A multilevel study of factors influencing 15-year-old middle school students’ suffering from school bullying” to “A multilevel analysis of factors influencing school bullying in 15-year-old middle school students

Line 2-3

Point 2: Abstract

Please refrain from using acronyms like PISA and ESCS in the abstract.

Response 2: We have added the full names to the abbreviations (PISA and ESCS).

Line 10-36

…Based on the database of four Chinese provinces and cities of the Program for International Student Assessment (PISA) in 2018 year, this study used a multilevel analysis model that combined school-level variables and student-level variables to explore the influencing factors of students' being bullied. Results: Students’ gender, grade repetition, truancy, and arriving late for class, economic, social, and cultural status (ESCS), …

Point 3: Introduction

The first entry line 28-30 is redundant, it can be cut.

Response 3: We have deleted the redundant sentence “school bullying has become a common concern of researchers all over the world, and”, so the new expression is “Nowadays, it is receiving more and more attention from many international organizations.”

Line 43-44

Point 4: Methods

I understand part of the data (individual variables) were obtained through self-report questionnaires filled out by students, whereas other school related variables were obtained by data provided by school principles. This should be clarified in the Materials section, in order to avoid potential confusion.

Response 4: Thanks for your kind reminder. We have added the information of school questionnaire to the Materials section.

Line 155-164

…First, we downloaded the 2018 global Student questionnaire data file and School questionnaire data file from the PISA website (https://www.oecd.org/pisa/data/2018database/). For a brief introduction to PISA and descriptions of the questionnaires, see the Appendix A. Then, we selected the data for mainland China. and the student questionnaire data of mainland China includes 12,058 middle school students aged 15 (from 15 years and 3 months to 16 years and 2 months), and the school questionnaire data includes 361 schools…

Point 5: Results

All the descriptive results should be presented succinctly in the Table 2 where min - max range should be added for continuous variables in order for mean levels to be read correctly and N, /% should be provided for categorical variables so that significant differences in proportions can be easily read. Reporting results correctly and comprehensively in Table 2 shall substitute all the text provided in the section 3.1 which is at this point redundant as it duplicates results already reported in the table.

Response 5: Thank you for the advice. To avoid duplication of information, we have merged table 1 and 2 into one table, and added min - max range for continuous variables. Because we converted all the category variables in the study into dummy variables, the Average refers to the percentage of the latter category in each variable, and the meaning of category variables were explained in the annotation of Table 1. Also, we have deleted section 3.1 to avoid duplicated results already reported in the table.

Line 192-195

For categorical variables, the Average refers to the percentage of the latter category in each variable. For example, for gender, 0.521 represents the proportion of male students is 52.1%; for Grade repetition, 0.063 represents the proportion of students’ repeating grade is 6.3%; and for School type, 0.140 represents the proportion of private schools is 14%. Other categorical variables are similarly interpreted.

Point 6: Discussion

Authors should start the discussion section 4.1 with their own findings rather than with findings of previous research. I suggest they move up lines 370 - 374 in the beginning and discuss how their results fit in or not with existing research.

The same applies to section 4.2 of the discussion.

Response 6: We have revised the discussion starting with our own findings, and move up lines 370 - 374 in the beginning to the second paragraph of the section 4.1. So as to the section 4.2.

Line 419-433

According to the results in Model II, in addition to grade and education type, the student-level variables of gender, grade repetition, truancy, and late for class all have significant positive effects on total school bullying and three types of bullying, while teacher support and parents’ support both have significant negative explanatory power on students' total school bullying and three types of bullying. ESCS only negatively affects students' total school bullying and physical bullying, not relational bullying and verbal bullying. The results above show that: boys suffer from a greater degree of school bullying than girls, and students who have repeated grades, truancy and are late for the class in the past two weeks are more severely bullied than those who have not. The lower the family's ESCS is, the higher the level of total school bullying and physical bullying. The lower the perceived teacher support and parents support are, the more severe the school bullying is. The above results are discussed further below.

First, in our study, we found that boys suffered more severe bullying than girls, both for total school bullying and for the three different types of bullying, which is partly consistent with previous studies…

Line 521-533

The results of Model III show that, only the school discipline atmosphere and the competitive atmosphere among students of the school environment level variables have significant impacts on the total school bullying and three types of bullying. School discipline atmosphere has a significant negative explanatory power to school bullying, while the competitive atmosphere among students has a significant positive explanatory power to students’ being bullied, indicating that students in schools with a less good discipline atmosphere experience greater levels of bullying than students in schools with a better one; students in schools with a high level of inter-student competition are more likely to experience higher levels of school bullying than those in schools with a lower level. A good school discipline atmosphere helps to protect students and make them less vulnerable to school bullying [23], but the competitive atmosphere among students may make some students feel jealous or hate of other classmates, which in turn increases the chances of students’ being bullied at school.

Line 473-474 calls for control analysis to be performed by authors. I suggest they either review this paragraph or perform additional analysis to test their interpretation of findings.

Response 6: We have reviewed this paragraph and we think perform additional analysis may be the focus of further research.

Line 574-577

…Due to the large number of control variables involved in our study, and different combinations of control variables will produce various different results, therefore, this part may need to be further explored in future studies.

Point 7 Limitations

It is not clear what the authors intend by “self-made questionnaires”? Please clarify. Also they state that “some factors were not related to school bullying” – please specify, which factors?

Response 7: Thank you for the questions. We have changed the “self-made questionnaires” into “questionnaires made by the researchers themselves”, hoping to be more clear about what we are referring to. And we enumerated some factors that “were not related to school bullying”, other results can be seen in the section 4.2.

Line 621-626

In follow-up research, other methods, such as using other databases or questionnaires made by the researchers themselves, should be used to include more influence factors to analyze this topic. In addition, some factors in our study were not obviously related to school bullying, such as grade and education type of student-level, and student behaviors and teacher behaviors that hindered learning, cooperative atmosphere among students of school-level, …

Point 8: Conclusions

The conclusion section should not be a dry repetition of the study results but it should contain take-home points based on the results and the potential impact that findings may have for research and practice. I suggest authors to review their conclusions from line 494 to line 512.

Response 8: Thank you for the recommendation. We have revised the conclusions which were contained not only basic results of our study but also corresponding measures that should be taken by all society, including teachers, family, school administrators, consulting teachers and other relevant personnel, etc.

Line 639-665

In the student-level variables, boys, students who have repeated grades, truancy and are late for the class in the past two weeks, and students whose economic, social, and cultural status is lower suffer from more severe school bullying. Besides, students who perceived lower teacher support and parents support are more severely bullied than those who have not. Thus, when developing school bullying interventions, more attention should be paid to these students. Teachers and parents should give more emotional sup-port to them. Also, school administrators, consulting teachers and other relevant personnel should pay more attention to students. Once they find signs of bullying, appropriate treatment should be provided in the first time to prevent the occurrence of the event.

In the school-level variables, students in schools with a less good discipline atmosphere and a higher level of inter-student competitive atmosphere experience greater levels of bullying. Therefore, when formulating plans to prevent school bullying at the school level, more consideration should be given to the important role of strengthening school disciplinary atmosphere and reducing the competitive atmosphere. A safety, well-ordered, positive and friendly school environment helps to protect students and make them less vulnerable to school bullying. And the establishment of such a campus environment needs the efforts of the whole society.
